# The Role of Mesothelin Expression in Serous Ovarian Carcinoma: Impacts on Diagnosis, Prognosis, and Therapeutic Targets

**DOI:** 10.3390/cancers14092283

**Published:** 2022-05-03

**Authors:** Giovanna Giordano, Elena Ferioli, Alessandro Tafuni

**Affiliations:** Pathology Unit, Department of Medicine and Surgery, University of Parma, Viale A. Gramsci, 14, 43126 Parma, Italy; ele.feri88@gmail.com (E.F.); alessandro.tafuni@unipr.it (A.T.)

**Keywords:** mesothelin, ovarian carcinoma, biomarker, mesothelin-targeting therapy

## Abstract

**Simple Summary:**

Ovarian cancer is the most lethal gynaecological malignancy, of which serous carcinoma is the most common subtype. The lack of symptoms and sensitive diagnostic tests for the early stages of its development may explain why diagnosis often occurs late, when the neoplasm has already spread outside the pelvis. Currently, the standard treatment for high-grade serous ovarian carcinomas (HSOCs) involves cytoreductive surgery followed by platinum-based systematic chemotherapy, which does not reduce either recurrence or mortality. Despite intense efforts to develop novel therapies using new chemotherapeutic agents, such as anti-angiogenesis agents and poly (ADP-ribose)-polymerase inhibitors, to improve patient outcomes, the five-year survival for this malignancy remains low. Therefore, it is important to identify new targetable molecules for the early diagnosis, monitoring, and treatment of this malignancy. The aim of this review is to discuss the role of mesothelin in serous ovarian carcinomas, focusing on diagnostic, prognostic, and therapeutic perspectives.

**Abstract:**

Mesothelin (MSLN) is a protein expressed in the mesothelial cell lining of the pleura, peritoneum, and pericardium; its biological functions in normal cells are still unknown. Experimental studies using knockout mice have suggested that this molecule does not play an important role in development and reproduction. In contrast, it has been observed that this molecule is produced in abnormal amounts in several malignant neoplasms, such as mesotheliomas and pancreatic adenocarcinomas. Many molecular studies have also demonstrated that mesothelin is overexpressed in HSOCs. Here, we discuss the current knowledge of mesothelin and focus on its role in clinical and pathological diagnoses, as well as its impact on the prognosis of HSOC. Moreover, regarding the binding of MSLN to the ovarian cancer antigen CA125, which has been demonstrated in many studies, we also report on signal transduction pathways that may play an important role in the spread and neoplastic progression of this lethal neoplasm. Given that mesothelin is overexpressed in many solid tumours and has antigenic properties, this molecule could be considered an antigenic target for the treatment of many malignancies. Consequently, we also review the literature to report on mesothelin-targeting therapies for HSOC that have been recently investigated in many clinical studies.

## 1. Introduction

Mesothelin (MSLN) is a glycoprotein located in the mesothelial lining of the body’s cavities and in many neoplasms [1]. It is anchored to the cell membrane by a glycosylphosphatidylinositol linkage. The mesothelin gene was first cloned by Chang and Pastan [1], and it encodes a precursor protein that is processed to yield a 40 kDa mesothelin protein and a 31 kDa soluble fragment. The human soluble fragment, named the megakaryocyte-potentiating factor (MPF), has been reported to have megakaryocyte-potentiating activity in mouse bone marrow [2]. In normal tissue, the physiological and biological functions of MSLN are still uncertain. Molecular biology studies demonstrated that a lack of MSLN in an MSLN knockout mouse model did not affect development, growth, or reproduction [3]. Conversely, MSLN is considered to be involved in several mechanisms of cancer pathogenesis. In ovarian carcinomas, it has been demonstrated that the binding of MSLN with its partner MUC16 (CA125) may play a role in cell adhesion, facilitating intra-peritoneal ovarian cancer metastasis [4,5,6].

There is evidence that mesothelin can be used as a new cancer biomarker [7] and as a target molecule for gene therapy [8]. Here, we discuss the current knowledge of MSLN, focusing on its role in clinical and pathological diagnoses, as well as its impact on the prognosis of HSOC. We also briefly examine the latest progress in mesothelin-targeting therapies for this aggressive and lethal neoplasm.

## 2. Mesothelin as a New Cancer Biomarker for the Diagnosis and Prognosis of Ovarian Carcinomas

Among gynaecological neoplasms, ovarian carcinomas have the highest mortality rates, since the diagnosis of this malignancy is often made late, occurring when the neoplasm is already at an advanced stage of development. The early detection of this type of neoplasm is difficult due to the absence of physical symptoms and the lack of sensitive screening methods [9].

Cancer antigen 125 (CA125) is currently the most commonly used serological biomarker for the diagnosis and management of patients with epithelial ovarian or fallopian tube or primary serous peritoneal cancers.

Many studies have suggested that CA125 can also be expressed at high levels in other types of cancers, such as breast cancer [10,11], mesotheliomas [12,13], non-Hodgkin’s lymphoma [14,15], and leukaemia [16], as well as leiomyomas and leiomyosarcomas of gastrointestinal origin [17]. CA125 was also found to be elevated in the sera of patients with such benign conditions as cirrhosis, ovarian cysts, endometriosis, pregnancy, congestive heart failure, and musculoskeletal inflammatory disorders [18].

In one study, only half of the studied patients with early-stage HSOC had elevated CA125 levels [19]. Thus, the sensitivity and specificity of CA125 for the detection of early-stage HSOC are, unfortunately, low [20]. Therefore, it is extremely important to identify new molecules for the early diagnosis and monitoring of this lethal neoplasm.

Concerning the use of MSLN as a biomarker for the diagnosis of HSOC, a significant amount of data in the literature suggest that this glycoprotein is expressed in different subtypes of ovarian carcinoma, especially HSOCs [21]. In one study, a splice variant of soluble mesothelin, named the soluble megakaryocyte-potentiating factor (SMRP), was found in the sera of patients with ovarian carcinoma [22]. Studies have reported that SMRP levels in the serum are significantly higher in subjects with HSOC than in either subjects with benign ovarian lesions or healthy subjects. It has also been observed that serum SMRP levels are related to the International Federation of Gynaecology and Obstetric (FIGO) system’s pathological staging and the grading of neoplasms, demonstrating that high serum levels of mesothelin may be indicative of tumour progression and poor survival [23,24,25].

Okla et al. observed that mesothelin levels in the peritoneal fluid did not differ significantly between patients with benign and malignant ovarian epithelial neoplasms. They also did not observe any differences in peritoneal fluid MSLN levels at different FIGO stages or between histological types of neoplasms. Thus, in contrast to the serum levels of MSLN, low levels of MSLN in the peritoneal fluid are not associated with a better prognosis [24].

Studies in the literature have reported that MSLN can also be detected in urine samples from patients affected by HSOC [26,27,28,29]. In particular, Badgwell et al. observed for the first time that urinary levels of MSLN could be considered to have greater sensitivity than serum levels in the early stages of HSOC [26]. Similarly, Hellstrom et al. demonstrated that in women with a pelvic mass, assaying urine for human epididymis protein 4 (HE4) or mesothelin could detect early HSOC more often than assaying the serum [27]. In their study, Hollevoet et al. demonstrated that mesothelin levels in the urine depended on impaired glomerular and tubular function, which could influence the interpretation of the mesothelin measurements and cause false-positive results [29]. Wu et al. considered SMRP serum levels to be promising markers for the diagnosis and monitoring of HSOC, but in combination with CA125 [23].

Since mesothelin is a membrane antigen that is overexpressed in a variety of solid neoplasms, including HSOC, there are many studies in the literature that have proven that radio immunoimaging analysis can be used for the non-invasive detection of MSLN-overexpressing tumours [30,31,32,33,34,35].

In several studies, anti-mesothelin antibodies were used and detected by fluorescence or magnetic resonance imaging [31,32]. For detection by positron emission tomography (PET), other authors have demonstrated that 89Zr-labelled antibodies can be used in ovarian models to target MSLN, forming an antibody–drug conjugate (ADC) that can provide information regarding both the organ distribution and drug dosing [32,33,34].

In diagnostic pathology, the immunohistochemical expression of MSLN could be used to distinguish between primary and metastatic ovarian carcinomas. In their paper, Kanner et al. demonstrated that MSLN expression could assist in differentiating Müllerian serous carcinomas from metastatic breast carcinomas (particularly those with a papillary morphology) and documented that none of the breast carcinomas were stained for MSLN [36]. Other studies have evaluated the expression of MSLN in HSOCs; the number of cases examined and the methods used are listed in Table 1. Ordóñez demonstrated that non-mucinous carcinomas of the ovary most frequently exhibited strong MSLN reactivity; however, they observed that this marker was also expressed in other non-mucinous carcinomas, such as clear-cell carcinomas of the ovary, endodermal sinus tumours, as well as clear-cell and transitional-cell carcinomas of the ovary [21]. While attempting to identify tumours that might benefit from targeted cancer therapies, Weidemann et al. observed that the highest prevalence of MSLN positivity was present in ovarian carcinomas (97% serous), based on analyses of tissue microarrays for MSLN expression in 122 different tumour types. Conversely, MSLN was rare in cancers of the breast, kidney, thyroid gland, soft tissues, and prostate [36].

The immunohistochemical expression of MSLN in the neoplastic section of HSOCs has also been investigated to establish its impact on prognosis. The literature provides limited and conflicting immunohistochemical data regarding MSLN expression and its prognostic impact on ovarian cancers. According to a study by Cheng et al., immunohistochemical MSLN expression was related to survival outcomes in patients with ovarian carcinomas. The authors observed that neoplasms with a high expression level of mesothelin showed a statistically worse prognosis than those with low immunoreactivity [37] (Table 1). Similarly, Yildiz et al. observed that a high expression level of MSLN in advanced serous ovarian cancers was associated with a poor prognosis and with worse platinum sensitivity in the advanced stage [38]. Cheng et al. observed that high MSLN expression, as investigated in a molecular study using real-time quantitative reverse-transcription polymerase chain reaction (PCR), was associated with chemo-resistance and poor survival in ovarian carcinomas [38] (Table 1). In contrast, while separating neoplasms with diffuse immunoreactivity from neoplasms with focal positivity, Yen et al. observed that diffuse MSLN expression was correlated with the prolonged survival of HSOC patients [39]. According to the authors, this finding could indicate that the immune response to mesothelin-expressing ovarian carcinoma cells may result in a reduction in tumour load and contribute to the patient’s prolonged overall survival.

Conversely, neoplasms with focal MSLN expression can progress given that neoplastic cells cannot be detected by the immune system and will continue to develop (Table 1). To validate the immunohistochemical results of eight frozen representative cases, Yen et al. used reverse-transcription PCR and observed that the PCR product of mesothelin was strongly representative of tumours with diffuse mesothelin immunoreactivity (4+ and 3+ positivity) (Figure 1A,B), while it was barely detectable in negative tumours (score: 0) (Table 1). The results reported by Yen et al. are not in accordance with those of other studies, in which a high expression level of MSLN has been associated with poor survival in other malignant epithelial neoplasms, such as lung adenocarcinomas and pancreatic ductal adenocarcinomas [40,41]. The conflicting data on MSLN expression and its prognostic impact on patients with ovarian carcinomas may be due to many factors, such as the different antibodies, protocols, and criteria used to evaluate immunoreactivity.

Magalhaes et al. conducted an immunohistochemical analysis and demonstrated that MSLN expression in patients with advanced serous carcinomas did not predict the clinical outcome but was correlated with CD11c+-positive immune infiltrate in neoplasms. MSLN expression was also significantly correlated with CD11c+ in the metastatic sites and in the perivascular areas of the primary neoplasm. Thus, the authors concluded that these data could also provide important information on the outcomes of immune-related therapies [42] (Table 1).

**Table 1 cancers-14-02283-t001:** Correlation between MSLN expression and prognosis in the main studies.

Authors	Case Number	Techinique Used for the Study	Prognosistic Data and Outcome
Ordóñez, N. [22]	14	Immunohistochemistry of formalin-fixed, paraffin-embedded neoplastic tissue	NR
Weidemann, S., et al. [36]	386	Immunohistochemistry of formalin-fixed, paraffin-embedded neoplastic TMA	No statistical association betwen MSLN expression and prognosis
Yildiz, Y., et al. [38]	42 advanced stage of HSOC	Immunohistochemistry of formalin-fixed, paraffin-embedded neoplastic tissue	High staining associated with platinum chemoresistence and worse OS
Cheng, et al. [37]	86	MSLN mRNA by RT-PCR on frozen neoplastic tissue	Positive MSLN expression correlated with chemoresistant and worse OS
Yen, M.J., et al. [39]	105 advanced stage of HSOC	Immunohistochemistry of formalin-fixed, paraffin-embedded neoplastic tissue and RT-PCR on eight cases	MSLN expression associated with prolonged survival
Magalhaes, I., et al. [42]	107	Immunohistochemistry of formalin-fixed, paraffin-embedded neoplastic TMA	No significant correaltion between positive MSLN and MSLN negative expression with OS in primary neoplasm and in the metastatic sites

HSOC: high-grade serous ovarian carcinoma; MSLN: mesothelin; NR: not reported; OS: overall survival; RT-PCR: real-time quantitative reverse-polymerase chain reaction. TMA: tissue microarrays.

Other studies have demonstrated that the expression of MSLN in the luminal membrane can be correlated with a worse prognosis than that associated with its cytoplasmic expression in gastric carcinoma, extrahepatic bile duct cancer, and breast cancer [43,44,45]. Kawamata et al. suggested that cytoplasmatic immunoreactivity is due to the presence of the 71 kDa precursor form, while luminal membrane staining likely indicates the presence of the 40 kDa membrane-bound form of MSLN, which represents an active form that is capable of promoting the aggressiveness of neoplasms by increasing cell motility, invasion, and growth in extrahepatic bile duct cancer [44].

To the best of our knowledge, there have been no immunohistochemical studies to date reporting a correlation between prognosis and the expression pattern of MSLN in HSOC. In addition, we did not find any studies that correlated the co-expression of MSLN and CA125 with prognosis in HSOC. However, in an immunohistochemical analysis of a cohort of 40 serous endometrial carcinoma cases, Kakimoto et al. observed that all 18 cases with the co-expression of these molecules had a worse prognosis compared to those without co-expression [46]. In our opinion, additional studies are necessary to elucidate whether the different patterns of MSLN immunoreactivity and co-expression with CA125 also have the same prognostic significance in HSOC in order to provide useful data to inform treatment procedures after surgical therapy.

Moreover, for a subset of occult HSOC, which has not arisen from the ovarian epithelium but from a lesion of the distal end of the fallopian tube, called serous tubal intraepithelial carcinoma (STIC), prophylactic salpingo-oophorectomy (PSO) is currently recommended to reduce the cancer risk and make an early diagnosis [47,48,49]. Clinically, it is very important to identify this subset of HSOC because complete surgery alone could cure affected patients, and even therapy using PARP inhibitors seems to be effective in cases at a high stage of development [50].

STIC appears to be a precursor lesion for pelvic (tubal, ovarian, or primary peritoneal) high-grade serous carcinoma (HGSC) and is a non-invasive subtype of HGSC, usually located at the distal fimbriated end of the fallopian tube, often related to BRCA1 or BRCA2 mutations and associated with breast cancer [48,49,50,51,52]. Upon pathological analysis, this lesion is extremely small and can be detected using a sectioning and extensively examining the fimbriated end (SEE-FIM) of the fallopian tube protocol and immunohistochemical analysis with specific antibodies, such as p53 and MIB-1 [53]. In the literature, we did not find any studies that evaluated the expression levels of MSLN in STIC and high-grade serous ovarian carcinoma (HSOC) arising from this lesion. Intuitively, one might think that since the tubal epithelium is columnar, unlike that of the mesothelium, the STIC might not express mesothelin, and consequently HSOCs that arise from STIC might also be negative for MSLN in immunohistochemical and molecular analyses.

However, we need to keep in mind that there is a broad range of tumour types, such as gastric, colorectal, oesophageal carcinomas, and synovial sarcoma, which although not arising from the mesothelium, can express MSLN [33]. In addition, as shown in Figure 1C, which refers to a case of STIC associated with small serous invasive carcinoma of the fimbria, both lesions showed marked positivity for MSLN upon immunohistochemical examination. Therefore, in our opinion, further studies with multiple cases of STIC, primary serous tubal carcinoma, and high-grade serous ovarian cancer associated with STIC or BRCA mutations should be investigated to clarify the role of MSLN in these malignancies and its impact on prognosis.

## 3. The Impacts of CA125, Other Molecules, and CA125–Mesothelin Binding on the Spread and Neoplastic Progression of HSOC

Many studies have demonstrated that both the mesothelial cells of the peritoneal lining and the neoplastic cells of HSOCs express mesothelin on their surfaces, and that the binding of mesothelin with MUC16/CA125 plays an important role in the spread and neoplastic progression of HSOC [4,5,6].

MUC16/CA125, present on the surfaces of neoplastic cells, can bind specifically to the mesothelin-expressing peritoneal lining due to its high affinity [6]. As a consequence, the MUC16/Ca125–MSLN link allows for the peritoneal implantation of ovarian neoplastic cells [54,55]. The adhesion of neoplastic cells to the peritoneal lining is also responsible for other subsequent events that characterise neoplastic progression, such as the invasion and diffusion of the neoplasm on the peritoneal surface and other organs.

Tumour cell detachment from the primary tumour, resulting in the diffusion of cancer cells into the peritoneal cavity, represents an early step of neoplastic diffusion related to an epithelial-to-mesenchymal transition (EMT). The loss of cell–cell adhesion is one of the most important and earliest steps of the EMT. This step is due to the dissolution of cell–cell junctions, and it is accompanied by unusual signalling events that cause a rearrangement of the cytoskeleton and the motile phenotype [55,56,57,58,59,60]. The loss of cell–cell adhesion and the acquisition of a fibroblast-like phenotype with migratory capabilities related to the EMT is due to the presence a proteinase family. These proteinases are produced by neoplastic cells and are named matrix metalloproteinases (MMPs). Type 1 matrix metalloproteinase (MMP-1) can both break down interstitial collagen, encouraging the invasion of neoplastic cells into the sub-mesothelial matrix, as well as catalyse the cleavage of CA125/MUC16 from the cell membrane [55]. The ectodomain shedding of CA125/MUC16, due to the catalytical action of MMP-1, has been supported by the immunohistochemical and molecular studies by Bruney et al. In their paper, an inverse relationship between MMP-1 and MUC16 expression in tissue sections of HSOCs following immunohistochemical analysis is reported. In fact, when neoplastic tissue showed strong immunoreactivity for MMP-1, it did not stain for CA125/MUC16. The authors observed similar results using cultured ovarian neoplastic cells (OVCAR3), which when engineered to overexpress MT1 (OVCAR433-MT), did not show MUC16 staining. On the contrary, when inactive mutant MT1-MPP was used, MUC16 immunoreactivity was restored. In addition, Bruney et al. observed the same changes in the expression of MUC16 mRNA in OVCA samples using real-time PCR analysis. The cleavage of CA125 allows for strong binding between ovarian neoplastic cells and the mesothelium through integrin-mediated adhesion and induces mesothelial cell retraction [55].

Conceivably, transforming growth factor ß (TGF ß) also represents another important molecule that promotes the EMT process. This molecule, produced by neoplastic cells, can induce the EMT process via the activation of other MMPs and is responsible for the loss of epithelial characteristics, including E-cadherin loss, as well as the invasion of neoplastic cells into the extra-cellular matrix [58]. After the binding of TGF ß with its receptor, which is present on the neoplastic cell membrane, SMAD1/2/3 molecules are released into the cytoplasm, where they link with SMAD4 to form a complex, which translated in the nucleus can bind to transcription factors, upregulating Snail 1/2. These molecules can both promote metalloproteinases and degrade E-cadherin, allowing for the loss of an epithelial phenotype of the neoplastic cell and the production of MMP3 and MMP9. The reduced expression of E-cadherin may lead to the loss of cell–cell adhesion and may result in cancer progression [59]. MMP-1 and MPP9, by degrading the basement membrane and IV collagen of the extra-cellular matrix, allow for the invasion and metastasis of the neoplasm [60]. These steps in the EMT process are supported by the study by Jin et al., who demonstrated an inverse relationship between Snail and E-cadherin expression upon immunohistochemical analysis using specific antibodies in a cohort of ovarian carcinomas. Interestingly, it was observed that a higher expression of Snail was present in the late stage of development of ovarian carcinoma and metastatic lesions than in the early tumours. Conversely, a higher expression level of E-cadherin was observed when Snail levels were low [59]. In addition, when E-cadherin was absent, Snail expression was elevated and localised in the nucleus. In this study, Jin et al. demonstrated via PCR study that the knockdown of Snail reduces the mRNA levels of MMPs [59].

More recently, in an in vitro study using OVCAR-3 ovarian cancer cell lines, Yuan et al. observed that the proliferation of ovarian cancer cells was not greatly affected by CA125; however, their migration increased with increasing concentrations of the substance. Moreover, the authors suggested that the effect of CA125 could be mediated via the Wnt signalling pathway; in fact, they observed that migration was inhibited by the Wnt antagonist Dickkopf-related protein 1 (DKK-1), while the DKK-1-mediated suppression of cell migration was reversed by CA125 [61].

Immunohistochemical studies have demonstrated that there are other molecules that can induce the EMT process in non-neoplastic diseases, such as endometriosis, and ovarian carcinomas. Furuya et al. observed that ZEB1 expression is a potential indicator of invasive endometriosis, as this can reduce E-cadherin expression [62]. Hosono et al. demonstrated that Twist represents other factors that can cause a reduction in E-cadherin expression, and this can be related to poor prognosis and increasing metastatic potential in ovarian carcinomas [63]. In addition, biological molecular studies have also proven that MicroRNA can play a role in the EMT of ovarian carcinoma [64]. MicroRNAs are small noncoding RNAs, which may function as oncogenes or tumour suppressors. Yang et al. demonstrated that there are MicroRNA-200 family members form a negative feedback loop and inhibit EMT-TFs of the Zeb family [64].

Wimberger et al., in their investigation, analysed the incidence and molecular phenotypes of EMT-like circulating tumour cells (CTCs) in the blood samples of ovarian cancer patients and monitored their responses to platinum-based chemotherapy, observing a selective enrichment of EMT-positive CTCs accompanied by the “de novo” emergence of dual PI3Kα- and Twist-positive CTCs, which may explain the therapy resistance [65].

In a more recent in vivo study, Hou et al. confirmed the important role of the binding of CA125 with mesothelin for the migration and metastatic diffusion of HOSCs [66]. Through their experiment in patient-derived xenograft studies, the authors proved that the binding of CA125 with mesothelin promotes the metastasis of ovarian cancer, and they hypothesised a cascade of molecular events involving many different molecules (Figure 2). The molecules that were suggested to play an important role in the diffusion and metastatic mechanism were serum/glucocorticoid-regulated kinase family 3 (SGK3), forkhead box O3 (FOXO3), and DKK-1. To validate the hypothesis that these factors were involved in the cascade, their levels were evaluated by performing a Western blot and semiquantitative analysis of the optical density after MSLN and CA125 binding. The authors found that free CA125 promoted ovarian cancer cell migration and tumour metastasis by binding with MSLN, which reduced DKK-1 expression and activated the SGK3/FOXO3 pathway (Figure 2).

SGK3 is a kinase that shares a similar structure and function with Akt; thus, it can phosphorylate FOXO3. After phosphorylation, FOXO3, by changing its conformation, binds with 14-3-3 proteins and is translated in the cytoplasm, preventing nuclear reimport (Figure 2) [67]. In the cytoplasm, FOXO3 is ubiquitinated and then degraded in a proteasome-dependent manner (Figure 2) [67]. These events represent an important step in carcinogenesis because FOXO3 is naturally located in the nucleus, where it regulates target genes such as p21 and p27, leading to cell cycle arrest and pro-apoptotic Bcl2-like protein11, Bim, promoting apoptosis to suppress tumourigenesis [68,69,70,71,72]. Consequently, the degradation of FOXO3 leads to increased proliferation and decreased apoptosis (Figure 2).

## 4. Mesothelin as a Therapeutic Target

Currently, ovarian cancer treatment consists of surgical tumour debulking complemented with taxane- and platinum-based chemotherapy [73], and occasionally supplemented with Avastin (bevacizumab, an antivascular endothelial growth factor therapy) [74]. In advanced or recurrent disease, or in patients with a BRCA mutation, maintenance therapy with a poly adenosine diphosphate (ADP-ribose) polymerase (PARP) inhibitor is an effective treatment option [75]. Thus, this subset of ovarian carcinoma is chemo-sensitive but a non-curable and indolent disease.

However, radical treatment regimens and multiple chemotherapeutic treatments do not reduce the recurrence of the disease nor the death rate of the patients. Given that MSLN is overexpressed in many solid tumours and has antigenic properties, this molecule could be considered an antigenic target for immunotherapeutic strategies in the treatment of ovarian carcinomas [76]. The main immunotherapeutic strategies that use different therapeutic agents include the anti-mesothelin immunotoxin SS1P, MORAb-009 (chimeric anti-mesothelin mAb), and the anti-mesothelin antibody–drug conjugate BAY-94 9343. Chimeric antigen receptor T cell (CAR T) therapy and vaccines have also been evaluated. The main immunotherapeutic strategies for ovarian cancer are summarised in Figure 3A. Table 2 lists the clinical trials of immunotherapeutic strategies for the treatment of ovarian carcinoma.

### 4.1. Anti-Mesothelin Immunotoxin SS1P

SS1P comprises an anti-MSLN immunotoxin obtained from immunised mice and fused to a truncated form of Pseudomonas exotoxin A (PE38) (Figure 3A). SS1P binding to MSLN forms a complex that is internalised by endocytosis and PE, translocated in the cytosol, and kills the cells that catalyse protein synthesis, thereby initiating programmed cell death [77] (Figure 3A). In vitro studies have demonstrated the cytotoxic effect of SS1P on the neoplastic cells of patients affected by ovarian carcinomas [77]. In a phase I clinical trial (ClinicalTrials.gov identifier: NCT00066651), patients with ovarian carcinomas presented with stable disease.

The side effects of the treatment are dose-related and include capillary leak syndrome and pleuritis due to SS1P binding to normal mesothelial cells, as well as inflammation. The association with prednisone reduces the risk of toxicity, allowing for increased dosages [78]. Moreover, in line with cases of mesothelioma, SS1P could be used in combination with chemotherapy to obtain a major response [78]. However, as observed in treatments for mesotheliomas, it must be kept in mind that the efficacy of SS1P is limited by anti-drug antibody formation. Thus, SS1P is being administered in association with pentostatin and cyclophosphamide, which are lymphocyte-depleting drugs that allow patients to receive multiple cycles of treatments [79].

### 4.2. MORAb-009 (Chimeric Anti-Mesothelin mAb)

MORAb-009 (chimeric anti-mesothelin mAb), also named amatuximab, is composed of the heavy- and light-chain variable regions of a mouse anti-mesothelin single-chain Fv grafted onto a human IgG1 and k constant region (Figure 3A) (Table 2). MORAb-009 has a high affinity for mesothelin, and a preclinical evaluation demonstrated that it could inhibit the adhesion between cell lines expressing mesothelin and MUC16 (CA125), as well as causing cell-mediated cytotoxicity in mesothelin-bearing tumour cells [80]. In clinical trials, it was observed that patients treated with MORAb-009 showed a marked increase in CA125 serum levels, suggesting that it could block the binding between mesothelin and CA125. It was also demonstrated that MORAb-009 could inhibit cellular adhesion during metastasis in the case of both ovarian carcinomas and mesotheliomas [81]. Studies in vivo on animal models demonstrated that these effects were markedly increased in combination with chemotherapy agents, such as gemcitabine and Taxol [82], or in a phase II clinical trial with other chemotherapeutic substances for cases of mesotheliomas (ClinicalTrials.gov identifier: NCT00738582) [83].

The reduction in the MPF level in serum after treatment demonstrated a correlation with good prognosis [83]. However, the combination with chemotherapy agents caused adverse events, such as hypersensitivity reactions, neutropenia, and atrial fibrillation [83].

Although most studies (ClinicalTrials.gov identifiers: NCT01521325, NCT01413451) on ovarian carcinomas have focused on the efficacy of monotherapy with MORAb-009, these data suggest that a combination with different chemotherapeutic agents could provide satisfactory results, with prolonged overall survival.

### 4.3. Anti-Mesothelin Antibody–Drug Conjugate (BAY-94 9343)

BAY-94 9343, known as anetumab ravtansine, is an anti-mesothelin antibody–drug conjugate (ADC) consisting of a fully human immunoglobulin G1 anti-mesothelin monoclonal antibody conjugated to the maytansine derivative tubulin inhibitor DM4 through a reducible disulphide linker (Figure 3A) [84]. BAY-94 9343 has antiproliferative activity because after binding to mesothelin on tumour cells it is internalised and the disulphide linker is cleaved, releasing DM4. Subsequently, DM4 binding to tubulin disrupts microtubule polymerisation, causing cell cycle arrest and apoptosis and consequently killing the dividing cells [85,86] (Figure 3A). Preclinical studies have shown that anetumab ravtansine is highly cytotoxic in MSLN-expressing mesotheliomas, as well as in pancreatic, non-small-cell lung, and ovarian cancer cell lines [83].

In an in vivo study, anetumab ravtansine was shown to have antitumour activity in mesotheliomas as well as in pancreatic and ovarian xenograft models [84]. A study by Quanz et al. demonstrated that in ovarian cancer cell lines and patient-derived xenografts, the combination of anetumab ravtansine with pegylated liposomal doxorubicin (PLD) or with carboplatin, copanlisib, or bevacizumab showed an additive antiproliferative activity both in vitro and in vivo compared to either agent used as a monotherapy [87].

### 4.4. Chimeric Antigen Receptor T cell (CAR T) Therapy

MSLN has also been regarded as an attractive target for chimeric antigen receptor T cell (CAR T) therapy because of its abundant expression in tumour cells and its limited expression in normal cells. CAR T therapy is a type of treatment in which a patient’s T cells, obtained by apheresis, are modified in the laboratory via the insertion of a gene for a special receptor called the chimeric antigen receptor (CAR). CAR T cells can target cell surface antigens without major histocompatibility complex (MHC) restriction. Thus, CAR T cells can be used for broad HLA-diverse allogeneic recipients.

The CAR is usually complex, with an extracellular antigen recognition domain that corresponds to a single-chain variable fragment (scFv) of a specific antibody, a transmembrane domain anchored to the cell membrane of the T cell, and an intracellular domain that transmits T cell activation signals. To amplify the activation signals in CARs, MSLN can be used in two co-stimulator domains, which allows for major activation in terms of proliferation, cytotoxicity, and consequently antitumour efficacy. The considerable effectiveness of this subtype of CAR, known as “*third-generation MSLN*”, has been proven in many neoplasms and in ovarian carcinoma [88] (Figure 3B). CAR T cells are grown in the laboratory and then administered to the patient by infusion. The CAR T cells are able to bind to antigens on the cancer cells and kill them. Once attached to the antigens present on the neoplastic cells, the CAR T cells become activated and stimulate the host’s immune system, which in turn attacks the MSLN-expressing cells [89]. The effectiveness of CAR T therapy has been observed in mouse models of different solid neoplasms, including ovarian carcinomas and mesotheliomas, in which the chimeric receptors recognised human MSLN and the inflammatory cytokines secreted by the T cells (including IL-2, IL-6, tumour necrosis factor alpha, and interferon-y) produced cytotoxic effects in the cancer cells (Figure 3B) (Table 2) [90,91]. Banville et al. provided insights into the design of logic-gated CAR T cell strategies with a greater number of antigens. The authors demonstrated that the most promising pairwise combination was CA125 and MSLN. Thus, a CAR T cell strategy against CA125 and MSLN would target most tumour cells in the majority of cases [92]. However, as observed in treatments for other neoplasms, it must be kept in mind that the immunosuppressive tumour microenvironment of neoplasms plays an important role in the response to CAR T therapy in vivo. Many authors have demonstrated that a transmembrane protein named programmed death ligand 1 (PD-L1) plays an important role in regulating the T cell response. The binding of this substance to the inhibitor programmed cell death protein 1 (PD-1) or the binding of PD-1 to the immune co-inhibitory receptor lymphocyte activation gene-3 (LAG3) transmits an inhibitory signal, causing a reduction in the proliferation of antigen-specific T cells, and consequently a reduction in the infiltration of T cells into the tumour lesion [93,94,95]. For ovarian carcinoma, recent in vivo preclinical studies have shown that it is possible to restore the functions of the tumour-specific checkpoint blockade in MSLN-directed CAR T cells using different substances [94,95,96].

The side effects of treatment observed during CAR T therapy are related to excessive immune activation, which causes cytokine release syndrome (CRS) and neurotoxicity. These adverse effects are probably due to non-specific T cell activation. CRS is an acute systemic inflammatory disorder characterised by fever, and sometimes by the fatal dysfunction of many organs [97,98]. Severe CRS symptoms can culminate in delirium, seizures, and encephalopathy caused by high levels of IL-6, IFN gamma, and CAR T cells in the cerebrospinal fluid [99]. Compartmental CRS (C-CRS) was reported in a patient with advanced ovarian cancer treated with mesothelin-targeted CAR T cells, characterised by the elevation of IL-6 and accumulation in the pleural fluid [98]. The treatment used against this serious side effect sometimes involves using an anti-IL-6R antibody, tocilizumab [100]. In cases with the involvement of the nervous system and unresponsive cases to tocilizumab, corticosteroids have often been used [99,100,101,102], or suicide genes have been introduced within T cells to reduce their number and activity (ClinicalTrials.gov identifiers: NTC0374965).

### 4.5. Vaccines

Cancer vaccines are immunotherapy treatments that induce a tumour-specific immune response in the host that is capable of recognising and eliminating neoplastic cells. The ability of T cells to recognise the antigens present on neoplastic cells and to produce an immune response capable of destroying them has long been known. In this type of immunotherapy for MSLN-positive cancers, Listeria monocytogenes, a Gram-positive bacterium, can be used as a vector. In humans, this bacterium causes infections with gastroenteritis, meningitis, and encephalitis; however, the human immune system is generally capable of controlling the disease [103,104]. The CRS-207 vaccine uses attenuated Listeria monocytogenes (Lm) (Lm ΔactA/ΔinlB) bacteria that are engineered to express human MSLN and can be used to treat MSLN-positive neoplasms (Figure 3B) (Table 2) [105].

The methods used to attenuate the virulence of Lm are mostly based on the deletion of certain genes that allow for sufficient infectivity and antigen production; however, it still has the potential to result in severe infection. Therefore, this treatment should be used with caution for patients with immunodeficiencies [106].

Treatment using CRS-207 with human-MSLN-expressing Listeria allows for the stimulation of the immune system with a robust response against neoplastic cells via different mechanisms. After fusion with a lysosome in the cytoplasm of an antigen-presenting cell, Lm can be killed; the secretions of its antigens into the cytosol as well as those prior to degradation in the phagosome can be loaded onto major histocompatibility complex (MHC) I and MCH II, causing the activation of potent CD 4 helper lymphocytes and CD 8 cytotoxic lymphocytes. In addition, during its entry into the antigen-presenting cell, Lm can activate proinflammatory genes through Toll-like receptors, which can amplify the response against neoplastic cells through the use of inflammatory cytokines [107] (Figure 3B) (Table 2).

## 5. Conclusions

In conclusion, the data collected from the literature suggest that MSLN can be used as a suitable marker for both the clinical and pathological diagnoses of HGSOC. Moreover, the typical expression pattern of MSLN in normal and cancer tissues makes it a promising target for therapeutic applications. Many studies have also suggested that a combination of mesothelin-targeting therapies with other different chemotherapeutic agents could provide satisfactory results, obtaining a major response and prolonged overall survival for patients affected by HGSOC. In addition, to prevent and reduce restrictions or side effects, such as anti-drug antibody formation or excessive immune activation, which can be observed in mesothelin-targeting therapies, other drugs can be administered.

In our opinion, although many clinical trials regarding MSLN-targeting therapies in ovarian carcinomas are ongoing, further studies will be useful for providing other satisfactory results and for establishing their effects on the health and behavioural outcomes of patients.

## Figures and Tables

**Figure 1 cancers-14-02283-f001:**
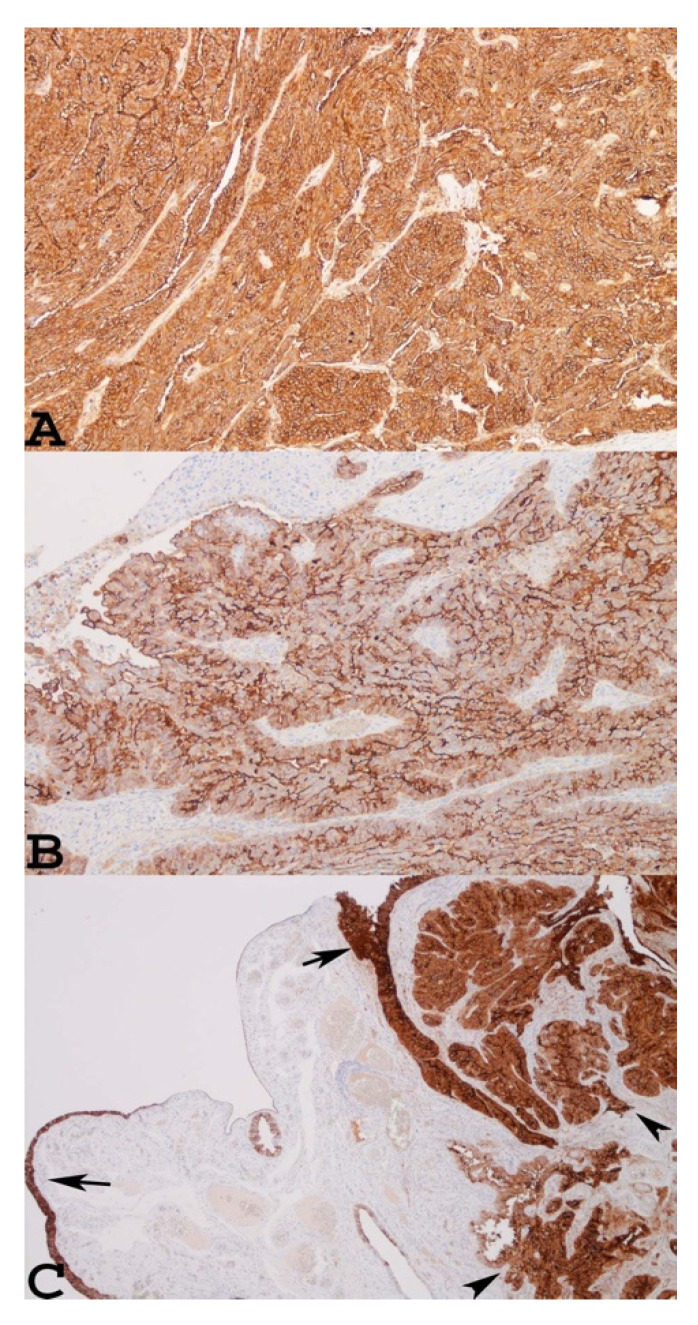
Two individual examples of advanced serous ovarian carcinoma showing diffuse mesothelin immunoreactivity: (**A**) score 4+, ×100; (**B**) score 3+, ×100. A personal example of STIC with a small nest of invasive serous carcinoma in a woman with BRCA-1 mutation and a previous history of breast cancer, showing strong immunoreactivity for MSLN in both lesions (**C**). Arrows indicate STIC, arrowheads indicate invasive serous carcinoma ×40.

**Figure 2 cancers-14-02283-f002:**
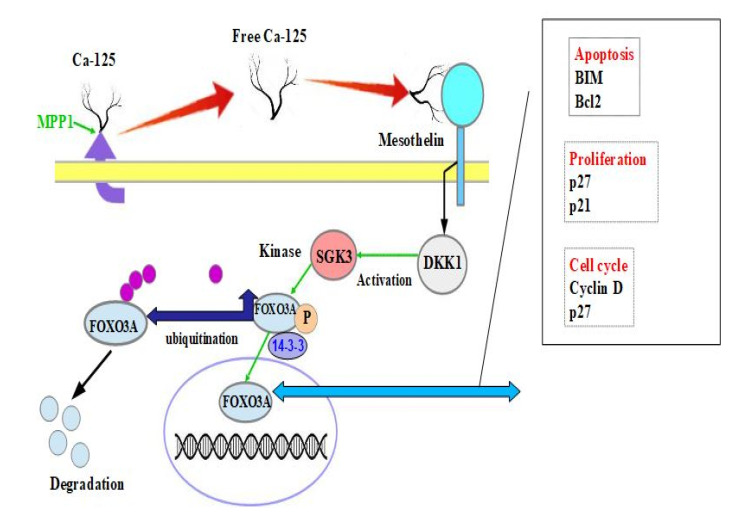
Schematic and simplified representation of the pathway involving the binding of CA125 with mesothelin for the migration of neoplastic cells and metastatic diffusion in advanced serous ovarian carcinoma.

**Figure 3 cancers-14-02283-f003:**
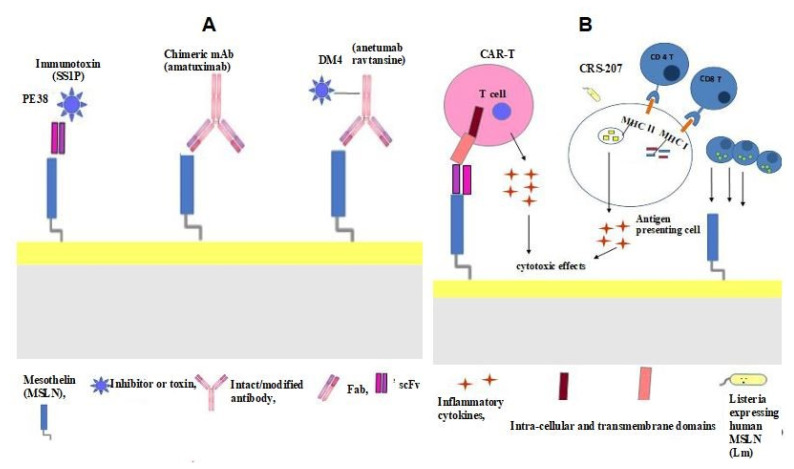
Schematic and simplified representation of the main therapeutic strategies that use mesothelin as a target. (**A**) PE translocated in cytosol and killed cells, catalysing protein synthesis and initiating programmed cell death. DM4 binding to tubulin disrupts the microtubule polymerisation, causing cell cycle arrest, apoptosis, and the killing of the dividing cells. (**B**) Attached to MSLN, CAR T cells become activated and stimulate the host immune system with the production of inflammatory cytokines. In CR-207, Listeria monocytogenes (Lm) and its antigens into the cytosol can be loaded onto major histocompatibility complex (MHC) I and MCH II, causing the activation of potent CD4 helper lymphocytes and CD8 cytotoxic lymphocytes, or activating pro-inflammatory genes, which can amplify the cytotoxic effect caused by inflammatory cytokines.

**Table 2 cancers-14-02283-t002:** Summary of clinical trials using MSLN target therapies for ovarian carcinoma.

Clinical Trials gov Identifier	Agent	Phase	Status	Disease Setting	Recruiting Centers
NCT00066651	SS1P	I	Completed	Advanced Cervical, ovarian Fallopian tube, pacreatic, peritoneal, lung, head and neck cancer	Unites States
NCT01521325	MORAb-009 (Chimeric Anti-Mesothelin mAb) amatuximab	I	Completed	Ovarian carcinoma, Mesothelioma, Pancreatic Cancer, Non Small Cell Lung	United States
NCT01413451	MORAb-009 (Chimeric Anti-Mesothelin mAb) amatuximab	Early Phase I	Terminated without efficacy in patienets with Ovarian Carcinoma	Ovarian carcinoma Mesothelioma, Pancreatic Cancer, Non Small Cell Lung cancer expressing mesothelin	United States
NCT01439152	BAY-94 9343 (Anti-MesothelinAntibody Drug Conjugate) Anetumab ravtansine	I	Completed	Invasive epithelial ovarian, primary serous peritoneal fallopian tube cancer	United States
NCT02751918	BAY-94 9343 (Anti-MesothelinAntibody Drug Conjugate) Anetumab ravtansine + pegyleted liposomal doxorubicin	Ib	Completed	Invasive or metastatic, predominantly epithelial platinum-resistant ovarian, fallopian tube, or primary serous peritoneal cancer	United States Belgium Moldova Spain
NCT03814447	CAR-T-meso	Early Phase I	Recruitment	Refractory-Relapsed Ovarian Cancer	China
NCT03608618	CAR-T-meso+ intraperitoneal MCY-M11	Early Phase I	Recruitment	Advanced Ovarian cancer and mesthelioma	United States
NCT00585845	CRS-207	I	Terminated	Ovarian Carcinoma, Mesothelioma, Non small-cell Lung carcinoma, Pancratic carcinoma, who have failed or who are not candidates for standard treatments	United States
NCT02575807	CRS-207 + alfaPD-1 + IDO1 inhibitor (Epacadostat)	I /II	Terminated low enrollment and lack of cinical activity	Platinum-resistant ovarian, fallopian or seous peritoneal cancer	United States

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
