# Peer review of "The Role of Mesothelin Expression in Serous Ovarian Carcinoma: Impacts on Diagnosis, Prognosis, and Therapeutic Targets"

_cancers, 2022, doi:10.3390/cancers14092283_

Round 1

Reviewer 1 Report

I agree with accepting this manuscript.

Page6 Line196 Kaimoto→Kakimoto

Author Response

FOR REVIEWER # 1

Comments and Suggestions for Authors
I agree with accepting this manuscript.

Page6 Line196 Kaimoto→Kakimoto

Re: Thank you very much for your positive comment

Kaimoto was replaced with Kakimoto (page: 7, Line:8).

Sincerely 
Have a nice day 

Giovanna Giordano

Reviewer 2 Report

The comments of the reviewers have mostly been addressed.  The list of clinical studies is now complemented by a structured table. Certain chapters are more concise than before.

However, taken together after  more than 15 years of studies, that were summarized,  the author should arrive at a final conclusion, whether Mesothelin is suitable as a target or prognostic marker and under which clnical conditions, for which patient group.

In addition, the request, to use the term high-grade serous carcinoma instead of serous carcinoma has not been applied consequently.  In the Abstract  and other places the term "serous ovarian carcinoma" is still used instead of high-grade serous carcinoma.  Please check thoroughly.

In addition, the paper still needs careful language editing by a science editor and native speaker.  I cannot start to point out individual terms or expressions , as I am not a native speaker myself.  But the overall flow of the paper suggests, that there is still significant potential for improvement.

Author Response

FOR REVIEWER # 2

The comments of the reviewers have mostly been addressed.  The list of clinical studies is now complemented by a structured table. Certain chapters are more concise than before.

Re: thank you very much for your comment and for suggestions to improve our paper

However, taken together after  more than 15 years of studies, that were summarized,  the author should arrive at a final conclusion, whether Mesothelin is suitable as a target or prognostic marker and under which clnical conditions, for which patient group.

Re: conclusion was modified: (page: 16)

In conclusion, the literature data collected suggest that MSL can be used as a suitable marker for both the clinical and pathological diagnosis of HGSOC. Moreover, a typical expressing pattern of MSLN in normal and cancer tissues makes it a promising target for therapeutic applications. Many studies have also suggested that a combination of mesothelin-targeting therapies with other different chemotherapeutic agents could provide satisfactory results, obtaining a major response and prolonged overall survival for patients affected by HGSOC. In addition, to prevent and reduce restrictions or side effects, such as anti-drug antibody formation or excessive immune activation, which can be observed in mesothelin-targeting therapies, other drugs can be administered.
In our opinion, although many clinical trials regarding MSLN-targeting therapies in ovarian carcinomas are ongoing, further studies will be useful for providing other satisfactory results and for establishing their effects on the health and behavioural outcomes of patients.

In addition, the request, to use the term high-grade serous carcinoma instead of serous carcinoma has not been applied consequently.  In the Abstract  and other places the term "serous ovarian carcinoma" is still used instead of high-grade serous carcinoma.  Please check thoroughly.

Re: the term serous ovarian carcinoma was replaced with high grade serous carcinoma (HSOC) in the abstract and in all sections of paper.

In addition, the paper still needs careful language editing by a science editor and native speaker.  I cannot start to point out individual terms or expressions , as I am not a native speaker myself.  But the overall flow of the paper suggests, that there is still significant potential for improvement.
Re:English language has been evaluated by native  English speakers, provided by the Journal service Please See  following certificates

Sincerely 
Have a nice day 

Giovanna Giordano

Reviewer 3 Report

The review by Giordano and co-authors is entitled "The Role of Mesothelin Expression in Serous Ovarian Carcinoma: Impact on Diagnosis, Prognosis, and Therapeutic Targets"

In this manuscript the authors attempted to characterize the role of the MSLN glycoprotein in the diagnosis and treatment strategies of ovarian carcinomae. There are several concerns that must be addressed by the authors:

The section "Mesothelin as a New OC Biomarker" contains scattered (often contradictory) data.

For example (lines 117-120): "the immunohistochemical expression of MSLN could be served to distinguish between primary and metastatic ovarian carcinomas. In their paper, Kanner et al. demonstrated that MSLN expression could assist in differentiating  Müllerian serous carcinomas from metastatic breast carcinomas". What neoplasms the authors speak about? The MSLN expression in BC differs from that in OC.

The section "Co-expression of mesothelin and CA125 in HSOC" is mainly about CA125, while mesothelin is far-fetched.

Where EMT is considered in relation to CA125, a whole paragraph regarding Snail EMT-TF is given, while there is not a word about other transcription factors regulating EMT (including those in the OC). At the same time, there are data in the literature concerning Zeb1 (Furuya, M., Masuda, H., Hara, K., Uchida, H., Sato, K., Sato, S., Asada, H., Maruyama, T., Yoshimura, Y., Katabuchi, H., Tanaka, M., & Saya, H. (2017). ZEB1 expression is a potential indicator of invasive endometriosis. Acta obstetricia et gynecologica Scandinavica96(9), 1128–1135. https://doi.org/10.1111/aogs.13179; Yang, J., Zhou, Y., Ng, S. K., Huang, K. C., Ni, X., Choi, P. W., Hasselblatt, K., Muto, M. G., Welch, W. R., Berkowitz, R. S., & Ng, S. W. (2017). Characterization of MicroRNA-200 pathway in ovarian cancer and serous intraepithelial carcinoma of fallopian tube. BMC cancer17(1), 422. https://doi.org/10.1186/s12885-017-3417-z) and Twist (Hosono, S., Kajiyama, H., Terauchi, M., Shibata, K., Ino, K., Nawa, A., & Kikkawa, F. (2007). Expression of Twist increases the risk for recurrence and for poor survival in epithelial ovarian carcinoma patients. British journal of cancer96(2), 314–320. https://doi.org/10.1038/sj.bjc.6603533; Wimberger, P., Hauch, S., Kimmig, R., & Kuhlmann, J. D. (2017). EMT-like circulating tumor cells in ovarian cancer patients are enriched by platinum-based chemotherapy. Oncotarget8(30), 48820–48831. https://doi.org/10.18632/oncotarget.16179) transcription factors.

The section entitled "Mesothelin as a therapeutic target" describes (lacking any discussion) known-to-date MSLN-based therapeutic strategies and restrictions/side effects of the treatments.

The Conclusions section is a set of general phrases and it raises more questions than answers.

And finally, authors need to do a spell check of their manuscript as there are some (often curious) mistakes throughout the text and negligence in manuscript preparation, so the manuscript must be improved to meet the criteria established in the journal's “Instructions for Authors”.

The reviewer cannot recommend this manuscript as a candidate for publication in Cancers journal in the current version.

Author Response

For REVIEWER # 3

Comments and Suggestions for Authors

The review by Giordano and co-authors is entitled "The Role of Mesothelin Expression in Serous Ovarian Carcinoma: Impact on Diagnosis, Prognosis, and Therapeutic Targets"

In this manuscript the authors attempted to characterize the role of the MSLN glycoprotein in the diagnosis and treatment strategies of ovarian carcinomae. There are several concerns that must be addressed by the authors:

The section "Mesothelin as a New OC Biomarker" contains scattered (often contradictory) data.

For example (lines 117-120): "the immunohistochemical expression of MSLN could be served to distinguish between primary and metastatic ovarian carcinomas. In their paper, Kanner et al. demonstrated that MSLN expression could assist in differentiating Müllerian serous carcinomas from metastatic breast carcinomas".

RE: Thank you for your observations

Contradictory sentences have been removed, moreover, We modified this sentence in The section "Mesothelin as a New OC Biomarker (page: 5)

In their paper, Kanner et al. demonstrated that MSLN expression could assist in differentiating Müllerian serous carcinomas from metastatic breast carcinomas (particularly those with a papillary morphology) and documented that none of the breast carcinomas was stained for mesothelin [38].

What neoplasms the authors speak about? The MSLN expression in BC differs from that in OC.

Re: in the same section about the expression of MSLN we have reported that there have been no immunohistochemical studies to date reporting a correlation between prognosis and the expression pattern of MSLN in HSOC (pages: 6-7)

Other studies have demonstrated that the expression of MSLN in the luminal membrane can be correlated with a worse prognosis than that associated with its cytoplasmic expression in gastric carcinoma, extrahepatic bile duct cancer, and breast cancer [45-47]. Kawamata et al. suggested that cytoplasmatic immunoreactivity is due to the presence of the 71 kDa precursor form, while luminal membrane staining likely indicates the presence of the 40 kDa membrane-bound form of MSLN, which represents an active form that is capable of promoting the aggressiveness of neoplasms by increasing cell motility, invasion, and growth in extrahepatic bile duct cancer [46].

The section "Co-expression of mesothelin and CA125 in HSOC" is mainly about CA125, while mesothelin is far-fetched.

Re: regarding Co-expression of mesothelin and CA125 in HSOC" SEE SECTION Mesothelin as a New Cancer

Biomarker for the Diagnosis and Prognosis of Ovarian Carcinomas (pages: 6-7)

To the best of our knowledge, there have been no immunohistochemical studies to date reporting a correlation between prognosis and the expression pattern of MSLN in HSOC. In addition, we did not find any studies that correlated the co-expression of MSLN and CA125 with prognosis in HSOC. However, in an immunohistochemical analysis of a cohort of 40 serous endometrial carcinoma cases, Kakimoto et al. observed that all 18 cases with the co-expression of these molecules had a worse prognosis compared to those without co-expression [48]. In our opinion, additional studies are necessary to elucidate whether the different patterns of MSLN immunoreactivity and co-expression with CA125 also have the same prognostic significance in HSOC, in order to provide useful data to inform treatment procedures after surgical therapy.

On contrary for other neoplasms there are studies which have demonstrated that the expression of MSLN in the luminal membrane can be correlated with a worse prognosis than that associated with its cytoplasmic expression in gastric carcinoma, extrahepatic bile duct cancer, and breast cancer [45-47]. Kawamata et al. suggested that cytoplasmatic immunoreactivity is due the presence of the 71 kDa precursor form, while luminal membrane staining likely indicates the presence of the 40 kDa membrane-bound form of MSLN, which represents an active form that is capable of promoting the aggressiveness of neoplasms by increasing cell motility, invasion, and growth in extrahepatic bile duct cancer [46].

THE TITLE "Co-expression of mesothelin and CA125 in HSOC" WAS REPLACED WITH MORE APPROPRIATE TITLE The impact of CA125, other molecules, and CA125-mesothelin binding on the spread and neoplastic progression of HSOC.

In this section we have reported the role of ca 125 in neoplastic progression and the molecular events due to interactions of ca 125 with MSLN ( page: 10, lines: 21-27; page 11, Lines: 1-11 and figure 2)

that could be responsible of neoplastic progression in HSOC

Where EMT is considered in relation to CA125, a whole paragraph regarding Snail EMT-TF is given, while there is not a word about other transcription factors regulating EMT (including those in the OC). At the same time, there are data in the literature concerning

RE Thank you for these suggestion

The data of following papers

Zeb 1 (Furuya, M., Masuda, H., Hara, K., Uchida, H., Sato, K., Sato, S., Asada, H., Maruyama, T., Yoshimura, Y., Katabuchi, H., Tanaka, M., & Saya, H. (2017). ZEB1 expression is a potential indicator of invasive endometriosis. Acta obstetricia et gynecologica Scandinavica, 96(9), 1128–1135.

https://doi.org/10.1111/aogs.13179;

Yang, J., Zhou, Y., Ng, S. K., Huang, K. C., Ni, X., Choi, P. W., Hasselblatt, K., Muto, M. G., Welch, W. R., Berkowitz, R. S., & Ng, S. W. (2017). Characterization of MicroRNA-200 pathway in ovarian cancer and serous intraepithelial carcinoma of fallopian tube. BMC cancer, 17(1), 422.

https://doi.org/10.1186/s12885-017-3417-z)

and

Twist (Hosono, S., Kajiyama, H., Terauchi, M., Shibata, K., Ino, K., Nawa, A., & Kikkawa, F. (2007). Expression of Twist increases the risk for recurrence and for poor survival in epithelial ovarian carcinoma patients. British journal of cancer, 96(2), 314–320. https://doi.org/10.1038/sj.bjc.6603533;

Wimberger, P., Hauch, S., Kimmig, R., & Kuhlmann, J. D. (2017). EMT-like circulating tumor cells in ovarian cancer patients are enriched by platinum-based chemotherapy. Oncotarget, 8(30), 48820–48831. https://doi.org/10.18632/oncotarget.16179)

transcription factors.

were added in the section  The impact of CA125, other molecules, and CA125-mesothelin binding on the

spread and neoplastic progression of HSOC. (Page 10)

Immunohistochemical studies have demonstrated that there are other molecules that can induce the EMT process in non-neoplastic diseases, such as endometriosis, and ovarian carcinomas. Furuya et al. observed that ZEB1 expression is a potential indicator of invasive endometriosis, as this can reduce E-Cadherin expression [65]. Hosono et al. demonstrated that Twist represents other factors that can cause a reduction of E-Cadherin expression, and this can be related to poor prognosis and increasing metastatic potential of ovarian carcinomas [66]. In addition, biological molecular studies have also proved that MicroRNA can play a role in the EMT of ovarian carcinoma [67]. MicroRNA are small noncoding RNA, which may function as oncogenes or tumour suppressors. Yang et al. demonstrated that there are MicroRNA-200 family members that are over-expressed in ovarian carcinomas and responsible for the upregulation of the E-Cadherin transcriptional repressor genes ZEB1, ZEB2, TGF ß1, and TGF ß2 [67].

Wimberger et al., in their investigation, analysed the incidence and molecular phenotypes of EMT-like circulating tumour cells (CTCs) in the blood of ovarian cancer patients and monitored their response to platinum-based chemotherapy, observing a selective enrichment of EMT-positive CTCs accompanied by the “de novo” emergence of dual PI3Kα and Twist positive CTCs, which may explain therapy resistance [68].”

The section entitled "Mesothelin as a therapeutic target" describes (lacking any discussion) known-to-date MSLN-based therapeutic strategies and restrictions/side effects of the treatments.

Re: thanks very much for this pertinent suggestions

We have added restrictions/side effects of the treatments observed in many studies and as well as related references were inserted (page: 12; lines: 8-15; page 12, lines: 25-28; page 13, Lines: 1-6; page: 14, Lines:27-28; page 15, Lines: 1-10)

Also see conclusion for discussion of therapeutic strategies Page 16

The Conclusions section is a set of general phrases and it raises more questions than answers.

Re: Conclusions was modified as: (page 16)

In conclusion, the literature data collected suggest that MSL can be used as a suitable marker for both the clinical and pathological diagnosis of HGSOC. Moreover, a typical expressing pattern of MSLN in normal and cancer tissues makes it a promising target for therapeutic applications. Many studies have also

suggested that a combination of mesothelin-targeting therapies with other different chemotherapeutic

agents could provide satisfactory results, obtaining a major response and prolonged overall survival for patients affected by HGSOC. In addition, to prevent and reduce restrictions or side effects, such as anti-drug antibody formation or excessive immune activation, which can be observed in mesothelin-targeting therapies, other drugs can be administered.

In our opinion, although many clinical trials regarding MSLN-targeting therapies in ovarian carcinomas are ongoing, further studies will be useful for providing other satisfactory results and for establishing their effects on the health and behavioural outcomes of patients.

And finally, authors need to do a spell check of their manuscript as there are some (often curious) mistakes throughout the text and negligence in manuscript preparation, so the manuscript must be improved to meet the criteria established in the journal's “Instructions for Authors”.

Re:English language has been evaluated by native English speakers, provided by the MDPI service Please See following certificates

Round 2

Reviewer 3 Report

The authors have improved their manuscript significantly, but some concerns shown below should be addressed by the authors before publication:

"MicroRNA-200 family members that are over-expressed in ovarian carcinomas and responsible for the upregulation of the E-Cadherin transcriptional repressor genes ZEB1, ZEB2, TGF ß1, and TGF ß2"

Dear authors, miR200 family members, on contrary, form a negative feedback loop and inhibit EMT-TFs of Zeb family. 

Citing Yang et al.: "Ovarian epithelial tumor cells showed concurrent up-regulation of miR-200, down-regulation of the four target genes (ZEB1, ZEB2, TGFβ1 and TGFβ2)"

The revised version of the paper is still full of mistakes and negligence in manuscript wrapping-up:

- L-6 instead of IL-6 (p.14)

- [Maude 2018] instead of [104] (p.14)

- In the Conclusions section MSL abbreviation for mesothelin used (instead of MSLN)

etc.

Author Response

For Reviewer #3

The authors have improved their manuscript significantly, but some concerns shown below should be addressed by the authors before publication

Re: thank you very much for your positive comment

"MicroRNA-200 family members that are over-expressed in ovarian carcinomas and responsible for the upregulation of the E-Cadherin transcriptional repressor genes ZEB1, ZEB2, TGF ß1, and TGF ß2"

Dear authors, miR200 family members, on contrary, form a negative feedback loop and inhibit EMT-TFs of Zeb family. 

Citing Yang et al.: "Ovarian epithelial tumor cells showed concurrent up-regulation of miR-200, down-regulation of the four target genes (ZEB1, ZEB2, TGFβ1 and TGFβ2)"

Re: we replace this sentence with: Yang et al. demonstrated that there are MicroRNA-200 family members form a negative feedback loop and inhibit EMT-TFs of Zeb family [67]. (p:10).

The revised version of the paper is still full of mistakes and negligence in manuscript wrapping-up:

- L-6 instead of IL-6 (p.14)

Re: we have replaced L-6 with IL-6 (p.15)

- [Maude 2018] instead of [104] (p.14)

Re:we have replaced Maude 2018 with ref 104 p 15

- In the Conclusions section MSL abbreviation for mesothelin used (instead of MSLN)

Re: we have replaced MSL with MSLN

Thank you for your time and your help to improve our manuscript.

Have a nice day 
G Giordano 

This manuscript is a resubmission of an earlier submission. The following is a list of the peer review reports and author responses from that submission.

Round 1

Reviewer 1 Report

This paper provides a state of the art, in depth systematic review of the role of Mesothelin in the tumorbiology and tumor microenvironment  of ovarian cancer.  Individual chapters describe Mesothelin as a New Cancer Biomarker in the diagnosis and prognosis of ovarian cancer and  Mesothelin as a new therapeutic target für directed antibodies and CAR T cell therapy.

The paper should be published, because it provides a good systematic, up to date review of the literature on Mesothelin in ovarian carcinoma. However, the following  issues should be addressed:

  1. The wealth of information sounds in several sections like a systematic collection of data and enumeration of the citations. Especially in the section of prognosis, the paper would benefit from table of evidence, summarizing the studies, with case number, technique of testing for mesothelin and the outcome.   Also, the chapter of immunotherapy would benefit from a more stringent and critical summary of the studies, rather than enumeration of the crucial findings.
  2. The term “serous ovarian carcinoma” should be replaced by “high-grade serous carcinoma” which is the commonly used term, unless the studies apply to both low-grade and high-grade serous carcinoma”.
  3. The Fig. 2 needs to be polished. Different letters and font are used throughout the Figure, which is confusing.  The authors should work with one font in different sizes and boldness. 

Author Response

Response to Reviewer #1: Manuscript Number: cancers-1561220

On behalf of my co-authors, I thank you for your time and effort to improve the paper and we hope that this revision is met satisfactorily. Changes have been highlighted at the particular points where it was revised.

Sincerely,

Giovanna Giordano

Corresponding author: Prof. Giovanna Giordano, MD, PhD, Department of Medicine and Surgery, University of Parma, Parma, Italy E-mail: giovanna.giordano@unipr.it

  1. 1. The wealth of information sounds in several sections like a systematic collection of data and enumeration of the citations. Especially in the section of prognosis, the paper would benefit from table of evidence, summarizing the studies, with case number, technique of testing for mesothelin and the outcome. Also, the chapter of immunotherapy would benefit from a more stringent and critical summary of the studies, rather than enumeration of the crucial findings.

Re: In the section of prognosis, we added table 1 to summarize the main studies in which mesothelin expression have been related with prognosis; for every study in this table was inserted number of cases considered, techniques used for testing mesothelin and the outcome. The section of immunotherapy was reduced

  1. 2. The term “serous ovarian carcinoma” should be replaced by “high-grade serous carcinoma” which is the commonly used term, unless the studies apply to both low-grade and high-grade serous carcinoma”.

  2. The term “serous ovarian carcinoma” was replaced by “high-grade serous carcinoma”
    3. The Fig. 2 needs to be polished. Different letters and font are used throughout the Figure, which is confusing. The authors should work with one font in different sizes and boldness.

  3. Re: The Fig 2 which currently corresponds to FIG 4, was polished, using one font with different sizes and boldness.

Reviewer 2 Report

This review article disscussed about the meaning of mesothelin expression and explained of the clinical studies of mesothelin targeting immunotherapy as a promissing therapeutic target molecule. It is very interesting and nice review. 

I want to add

  1. the curent topics of co-expression mesothelin and CA125, clinical and biological function.
  2. the biological function of mesothelin expression, membrane and cytoplasmic expression
  3. how to evaluate of mesothelin expression in each clinical study.

Author Response

Response to Reviewer #2: Manuscript Number: cancers-1561220

On behalf of my co-authors, I thank you for your time and effort to improve the paper and we hope that this revision is met satisfactorily. Changes have been highlighted at the particular points where it was revised.

Sincerely,

Giovanna Giordano

Corresponding author: Prof. Giovanna Giordano, MD, PhD, Department of Medicine and Surgery, University of Parma, Parma, Italy E-mail: giovanna.giordano@unipr.it

Manuscript Number: cancers-1561220

This review article disscussed about the meaning of mesothelin expression and explained of the clinical studies of mesothelin targeting immunotherapy as a promissing therapeutic target molecule. It is very interesting and nice review.

I want to add

1 the curent topics of co-expression mesothelin and CA125, clinical and biological function.

Re: The topics of co-expression mesothelin and Ca-125 with clinical and biological function were added in a new Chapter, entitled: “Co-expression of mesothelin and CA125 in HOSC, impact on spread and neoplastic progression”. (pag 9 Lines: 13-26, pag 10, Lines: 1-28)

Moreover, we have inserted a new figures (Figure 2-3) , to elucidate the role of these molecules in all phases of neoplastic progression of high grade of serous carcinoma.

2.the biological function of mesothelin expression, membrane and cytoplasmic expression

Re: The biological function of mesothelin expression as membranous and cytoplasmic expression was added in the chapter “Mesothelin Detection in Neoplastic Tissue” (pag. 8 , Lines: 21-26)

3 how to evaluate of mesothelin expression in each clinical study.

Re:The techniques used to evaluate mesothelin expression were listed in table 1

Reviewer 3 Report

Giordano et al review titled “The role of mesothelin expression in serious ovarian carcinoma: Impact on diagnosis, prognosis, and therapeutic target” explain prognostic, diagnostic and therapeutic potential of mesothelin in ovarian carcinoma. There are several in depth reviews written on this topic. I do not see any additional information/s in this review that would be of use to readers. 

  Insights Into the Role of Mesothelin as a Diagnostic and Therapeutic Target in Ovarian Carcinoma. Shen J, Sun X, Zhou J.Front Oncol. 2020 Aug 28;10:1263. doi: 10.3389/fonc.2020.01263. eCollection 2020.PMID: 32983962 Free PMC article. Review. Some suggestions 1.Though the article was submitted as review, I see material methods, results and other details. When you resubmit somewhere please remove that, and if you have some of your data include it as figures (personal communication).   2.Also in the abstract, try to avoid too much details about mesothelin rather try summarising what readers could glean from the review.    3. Include some tables showing the progress of mesothelin as a therapeutic target focusing only on Ovarian carcinoma and details preclinical, clinical phases of various trials.   4. Add some figures detailing the effort of understanding the function of mesothelin including signal transduction pathways again focusing only ovarian carcinoma        

Author Response

Response to Reviewer #3: Manuscript Number: cancers-1561220

On behalf of my co-authors, I thank you for your time and effort to improve the paper and we hope that this revision is met satisfactorily. Changes have been highlighted at the particular points where it was revised.

Sincerely,

Giovanna Giordano

Corresponding author: Prof. Giovanna Giordano, MD, PhD, Department of Medicine and Surgery, University of Parma, Parma, Italy E-mail: giovanna.giordano@unipr.it

Reviewer #3: Manuscript Number: cancers-1561220

1.Though the article was submitted as review, I see material methods, results and other details. When you resubmit somewhere please remove that, and if you have some of your data include it as figures (personal communication).

Re: Materials, methods and results were removed. In legend of FIG 1 we pointed out that examples documented were Two personal examples of high grade serous carcinoma showing diffuse mesothelin immunoreactivity.

  1. Also in the abstract, try to avoid too much details about mesothelin rather try summarising what readers could glean from the review.

Re: In the abstract, we removed some details about mesothelin and inserted more data which have emerged from our literature review. (pag 2, Lines: 13-27 and pag 3, Lines: 1-2)

  1. Include some tables showing the progress of mesothelin as a therapeutic target focusing only on Ovarian carcinoma and details preclinical, clinical phases of various trials

Re: We added Table 2 which summarizes the main Clinical Trials

4. Add some figures detailing the effort of understanding the function of mesothelin incudinig signal transduction pathways again focusing only ovarian carcinoma

Re: we added two new figures and new chapter entitled “ Co-expression of mesothelin and CA125 in HOSC, impact on spread and neoplastic progression”. (pag 9 Lines: 13-26, pag 10, Lines: 1-28) (FIG 2 AND FIG 3)

Fig 2 elucidates function of the mesothelin and Ca 125 in all phases of neoplastic progression, Fig 3 shows more recent knowledges about signal transduction pathways involving the mesothelin in neoplastic progression in cases of high serous ovarian carcinoma

Round 2

Reviewer 3 Report

Authors have definitely improved the review from the last one. However I would like author including following suggestion.

I can consider the review if the authors could include the following

  1. Author suggested EMT due to Mesothelin/MUC16 interaction. However there is no insight into how does happen?
  2. Has anyone quantitated the number of moleculss of mesothelin in different samples either from your lab or any other lab. Is there any RNA seq data available from large analysis using serious OC and could you please show a bar diagram showing the levels of expression from normal level that seen in peritoneum. Within the samples, one may expect low, medium and higher expression and what percentage of such analysis shows low, medium and high expression. Such a graph is highly encouraged.
  3. Fig 2. Is a good addition. However I would make it more legible. Bring the legend at the bottom. No need to write the word legend there. Also show like a flow chart and show what happens above the arrow. It is easy to understand that way. I strongly recommend to make a box at the EMT area of the figure and extend the figure to include a    potential signal transduction pathway that may support EMT as a continuation of figure. This necessarily need not be from SOC but could be from any cancer type and author could suggest that may happen in SOC and need to be investigated.
  4. Figure 3. Could you please write what enzyme cleave the Ca125 and where does it binds ( make it more easy to visualize) and also show an arrow to suggest down regulation of DKK1, activation of SGK3 and phosphorylation of FOX3A. Also who few targets of FOXO3A that that may regulate migration, apoptosis and cell cycle regulation as separate three arrows. Correct spelling mistake not stimolation I think stimulation. It is better to show FOXO3A switching from Non to  phosphorylated instead.
  5. Fig 5. Bring legend at the bottom, no need to write the word legend. Again the figure looks very cluttered. Is there any way you can make two figures Figure 4 A with A, B and C and Fig 4 B with DE and you could add additional strategies here. Also avoid writing at the bottom of the figure regarding mechanism rather show arrow and show the effect.
  6. I need a conclusion and future direction that is not only summarizing the write up but how this field can be moved further discussing the drawbacks, strengths and improvement need to be included.

Minor

Introduction was divided into many small paragraphs. Please bring this as a single cohesive paragraph. This tendency was seen throughout the review. Please don’t write one or two sentences as a separate  paragraph;

 Experimental studies using knockout mice suggest that this substance does not play an important role 22 in development and reproduction. I

There is evidence that mesothelin can be used as a new cancer biomarker [7] and as 52 a target molecule for gene therapy [8]

Here, we discuss the current knowledge of MSLN, focusing on its role in clinical and 54 pathological diagnoses as well as its impact on the prognosis of HSOC. We also briefly 55 examine the latest progress in mesothelin-targeting therapies for this aggressive and lethal 56 neoplasm.

Mesothelin as a New Cancer Biomarker for the Diagnosis and Prognosis of Ovarian 61 Carcinomas

This pattern Is  evident  everywhere. Summarise all relevant related info as a tight paragraph instead of writing as single or two sentences,

Please check the spelling mistake and grammar, expression wherever it is appropriate.

For example,

using knockout mice suggest that this substance does not play an important role 2 use other word instead of substance

Author Response

Authors have definitely improved the review from the last one. However I would like author including following suggestion.

I can consider the review if the authors could include the following

Author suggested EMT due to Mesothelin/MUC16 interaction. However there is no insight into how does happen?

Has anyone quantitated the number of moleculss of mesothelin in different samples either from your lab or any other lab. Is there any RNA seq data available from large analysis using serious OC and could you please show a bar diagram showing the levels of expression from normal level that seen in peritoneum. Within the samples, one may expect low, medium and higher expression and what percentage of such analysis shows low, medium and high expression. Such a graph is highly encouraged.

RE: The interaction of Mesothelin/MUC16 have reported in the chapter entitled “Co-expression of mesothelin and CA125 in HSOC: impact on spread and neoplastic progression.”As well as the molecules and mechanisms responsible of EMT have been described in the same Chapter, in which we have reported quantitated comparative evaluations for these molecules in studies made by other Researchers.

Bruney, L.; Conley, K.C.; Moss, N.M.; Liu, Y.; Stack, M.S. Membrane-type I matrix metalloproteinase-dependent ectodomain shedding of mucin16/CA-125 on ovarian cancer cells modulates adhesion and invasion of peritoneal mesothelium. Biol Chem. 2014, 395, 1221-1231. DOI: 10.1515/hsz-2014-0155.

Moreover, additional references were inserted.

55 Xu, J.; Lamouille, S.; Derynck,  R. TGF-beta-induced epithelial to mesenchymal transition. Cell Res. 2009, 19, 156-172. DOI: 10.1038/cr.2009.5.

56 Jin, H.; Yu, Y.; Zhang, T.; Zhou, X.; Zhou, J.; Jia, L.; Wu, Y.; Zhou, B.P.; Feng, Y. Snail is critical for tumor growth and metastasis of ovarian carcinoma. Int J Cancer. 2010, 126, 2102-11. DOI: 10.1002/ijc.24901.

57 Carey, P.; Low, E.; Harper, E.; Stack, M.S. Metalloproteinases in Ovarian Cancer. Int J Mol Sci. 2021, 22, 3403. DOI: 10.3390/ijms22073403.

Fig 2. Is a good addition. However I would make it more legible. Bring the legend at the bottom. No need to write the word legend there. Also show like a flow chart and show what happens above the arrow. It is easy to understand that way. I strongly recommend to make a box at the EMT area of the figure and extend the figure to include a  potential signal transduction pathway that may support EMT as a continuation of figure. This necessarily need not be from SOC but could be from any cancer type and author could suggest that may happen in SOC and need to be investigated.

RE: In FGURE 2 we have bringed the legend at the bottom. The word “legend “was removed.

We added a box to elucidate  potential signal transduction pathway that may support EMT in ovarian carcinoma, which also is explained the text

A box in the figure 2 was added   to include a  potential signal transduction pathway that may explained EMT

Figure 3. Could you please write what enzyme cleave the Ca125 and where does it binds ( make it more easy to visualize) and also show an arrow to suggest down regulation of DKK1, activation of SGK3 and phosphorylation of FOX3A. Also who few targets of FOXO3A that that may regulate migration, apoptosis and cell cycle regulation as separate three arrows. Correct spelling mistake not stimolation I think stimulation. It is better to show FOXO3A switching from Non to  phosphorylated instead.

Type 1 matrix metalloproteinase (MMP-1) both can break down interstitial collagen encouraging the invasion of neoplastic cells into the sub-mesothelial matrix and can catalise cleavage of  CA125/MUC16  from the cell membrane [51] (Green arrow in the FIG 2).

Regarding changes responsible of profiferation and apoptosis they  have been explained in the text and Figure 3.

Also, we added  the following references which can explain the role of Phosphorylated FOXO3A in the apoptosis, proliferation and cell cycle.

60 Liu, Y.; Ao, X.; Ding, W.; Ponnusamy, M.; Wu, W.; Hao, X.; Yu, W.; Wang, Y.; Li, P; Wang, J. Critical role of FOXO3a in carcinogenesis. Mol Cancer. 2018, 17: 104. DOI: 10.1186/s12943-018-08563

61 Chen YF, Pandey S, Day CH, Chen YF, Jiang AZ, Ho TJ, Chen RJ, PadmaViswanadha V, Kuo WW, Huang CY. Synergistic effect of HIF-1alpha and FoxO3a trigger cardiomyocyte apoptosis under hyperglycemic ischemia condition. J Cell Physiol. 2017;233(4):3660–71.

62 Gilley J, Coffer PJ, Ham J. FOXO transcription factors directly activate bim gene expression and promote apoptosis in sympathetic neurons. J Cell Biol. 2003;162(4):613-622. doi:10.1083/jcb.200303026

63 McClelland Descalzo DL, Satoorian TS, Walker LM, Sparks NR, Pulyanina PY, Zur Nieden NI. Glucose-induced oxidative stress reduces proliferation in embryonic stem cells via FOXO3A/beta-catenin-dependent transcription of p21(cip1). Stem Cell Reports. 2016;7(1):55–68.

64 McGowan SE, McCoy DM. Platelet-derived growth factor-a regulates lung fibroblast S-phase entry through p27(kip1) and FoxO3a. Respir Res. 2013;14:68.

65 Joseph J, Ametepe ES, Haribabu N, Agbayani G, Krishnan L, Blais A, Sad S. Inhibition of ROS and upregulation of inflammatory cytokines by FoxO3a promotes survival against Salmonella typhimurium. Nat Commun. 2016;7:12748.

A  red arrow that suggests down regulation of DKK1 was added, as well as green arrows were added to indicate activation of SGK3 and phosphorylation of FOX3A

The word stimolation has been replaced with stimulation.

Phosphorylated FOXO3A was graphically indicated in the figure 3

Fig 5. Bring legend at the bottom, no need to write the word legend. Again the figure looks very cluttered. Is there any way you can make two figures Figure 4 A with A, B and C and Fig 4 B with DE and you could add additional strategies here. Also avoid writing at the bottom of the figure regarding mechanism rather show arrow and show the effect.

There is not Figure 5, but figure 4, which now shows  two parts: 4A and 4B as you have suggested. The mechanisms have been removed and iserted in the text in the section “ Figure Legends”

I need a conclusion and future direction that is not only summarizing the write up but how this field can be moved further discussing the drawbacks, strengths and improvement need to be included.

to the Conclusion were added and future Perspectives

Minor

Introduction was divided into many small paragraphs. Please bring this as a single cohesive paragraph. This tendency was seen throughout the review. Please don’t write one or two sentences as a separate  paragraph;

Experimental studies using knockout mice suggest that this substance does not play an important role 22 in development and reproduction. I

There is evidence that mesothelin can be used as a new cancer biomarker [7] and as 52 a target molecule for gene therapy [8]

Here, we discuss the current knowledge of MSLN, focusing on its role in clinical and 54 pathological diagnoses as well as its impact on the prognosis of HSOC. We also briefly 55 examine the latest progress in mesothelin-targeting therapies for this aggressive and lethal 56 neoplasm.

Mesothelin as a New Cancer Biomarker for the Diagnosis and Prognosis of Ovarian 61 Carcinomas

This pattern Is  evident  everywhere. Summarise all relevant related info as a tight paragraph instead of writing as single or two sentences

Please check the spelling mistake and grammar, expression wherever it is appropriate.

For example,

using knockout mice suggest that this substance does not play an important role 2 use other word instead of substance

Introduction is a single cohesive paragraph, as well as entire document does not show separating sentences.

English language has been evaluated by native  English speakers, provided by the Journal service, as well as in all our previous submitted papers.

Please See  certificates in the attached file

Round 3

Reviewer 3 Report

Authors have improved the manuscript,  however, would like to see the following changes

  1. Still there are plenty of short paragraphs with one or two sentences and I strongly request authors to bring these things together wherever possible to make a good solid paragraph.

For examples

Among gynaecological neoplasms, ovarian carcinomas have the highest mortality because the diagnosis of this malignancy is often made late, occurring when the neo[1]plasm is already at an advanced stage of development.

 The early detection of this type of neoplasm is difficult due to the absence of physi[1]cal symptoms and the lack of sensitive screening methods [9].

In one study, only half of the studied patients with early-stage ovarian carcinomas had elevated CA125 levels [19].

Thus, the sensitivity and specificity of CA125 for the detection of early-stage ovari[1]an carcinomas are unfortunately low [20]. Therefore, it is extremely important to identify new molecules for the early diagnosis and monitoring of this lethal neoplasm

Avoid making paragraph out of one or two sentences. Please avoid these throughout the manuscript.

  1. Delete the following paragraph from the manuscript (page 3)

Radioimmunoimaging is a type of molecular nuclear medicine imaging that applies specific antibodies targeting tumour-specific antigens labelled with radionuclides for imaging [36]. Thus, this molecular imaging technique allows for the assessment of tu[1]mour uptake and distribution in the primary and secondary tumour sites, as well as the response to therapy. Different anti-mesothelin antibodies have been used in animal models.

  1. Is there any studies out there relating TGF beta, snail and mesothelin oherwise the figure is too speculative and need to be removed.
  2. page 3. In several studies, anti-mesothelin antibodies have been used and detected by fluorescence imaging or magnetic resonance

Change this into fluorescence or magnetic  resonance imaging

  1. Through their experiment In tumor xenograft mice change to patient derived xenograft studies revealed CA125…..

There are several such expression problems throughout the manuscript and please carefully go through and correct wherever possible.

  1. Figure 3. Where FOXO3 degradation is misleading and please remove that and also my understanding is that authors want to convey DKK1 is an inhibitor of SGK3and its absence activate SGK3. If that is the case put an arrow from Mesothelin to DKK and show a downward arrow near DKK showing the down regulation and show SGK3 as activated SGK3 kinase.

Author Response

For  Reviewer #3

Thank you very much for your additional  advices and recommendations to improve our manuscript

Authors have improved the manuscript,  however, would like to see the following changes

Still there are plenty of short paragraphs with one or two sentences and I strongly request authors to bring these things together wherever possible to make a good solid paragraph.

For examples

Among gynaecological neoplasms, ovarian carcinomas have the highest mortality because the diagnosis of this malignancy is often made late, occurring when the neo[1]plasm is already at an advanced stage of development.

 The early detection of this type of neoplasm is difficult due to the absence of physi[1]cal symptoms and the lack of sensitive screening methods [9].

In one study, only half of the studied patients with early-stage ovarian carcinomas had elevated CA125 levels [19].

Thus, the sensitivity and specificity of CA125 for the detection of early-stage ovari[1]an carcinomas are unfortunately low [20]. Therefore, it is extremely important to identify new molecules for the early diagnosis and monitoring of this lethal neoplasm

Avoid making paragraph out of one or two sentences. Please avoid these throughout the manuscript.

RE: Throughout  entire manuscript several paragraphs have been reduced. Thank you

Delete the following paragraph from the manuscript (page 3)

Radioimmunoimaging is a type of molecular nuclear medicine imaging that applies specific antibodies targeting tumour-specific antigens labelled with radionuclides for imaging [36]. Thus, this molecular imaging technique allows for the assessment of tu[1]mour uptake and distribution in the primary and secondary tumour sites, as well as the response to therapy. Different anti-mesothelin antibodies have been used in animal models.

RE: This  paragraph has been deleted, as well as  the reference # [36]

Is there any studies out there relating TGF beta, snail and mesothelin oherwise the figure is too speculative and need to be removed.

Re: in our opinion  this figure could be inserted because

In the box  was reported a probable pathway responsible of epithelial-mesenchymal transition (EMT) in ovarian carcinoma, which represents an important step in the neoplastic progression that follows the adhesion of mesothelin with ca 125.

 I insert this because in  previous reviewer comment there were these advices:

Fig 2. Is a good addition. However I would make it more legible. Bring the legend at the bottom. No need to write the word legend there. Also show like a flow chart and show what happens above the arrow. It is easy to understand that way. I strongly recommend to make a box at the EMT area of the figure and extend the figure to include a  potential signal transduction pathway that may support EMT as a continuation of figure. This necessarily need not be from SOC but could be from any cancer type and author could suggest that may happen in SOC and need to be investigated.

RE: In FGURE 2 we have bringed the legend at the bottom. The word “legend “was removed.

We added a box to elucidate  potential signal transduction pathway that may support EMT in ovarian carcinoma, which also is explained the text

A box in the figure 2 was added   to include a  potential signal transduction pathway that may explained EMT

However, I removed it, as you have suggested

page 3. In several studies, anti-mesothelin antibodies have been used and detected by fluorescence imaging or magnetic resonance

Change this into fluorescence or magnetic  resonance imaging

Re, The sentence has been modified:

 In several studies, anti-mesothelin antibodies have been used and detected by fluorescence  or magnetic resonance imaging [31,32].

Through their experiment In tumor xenograft mice change to patient derived xenograft studies revealed CA125…..

Re: sentence has been modified as Through their experiment in patient derived xenograft studies, the authors proved that CA125 promotes the metastasis of ovarian cancer, and they hypothesised a cascade of molecular events involving many different molecules [FIG. 32].

There are several such expression problems throughout the manuscript and please carefully go through and correct wherever possible.

Re: English language was been evaluated and corrected several times by native speakers of MDPI service provided by journal .

Howerver, some sentences were modified

Figure 3. Where FOXO3 degradation is misleading and please remove that and also my understanding is that authors want to convey DKK1 is an inhibitor of SGK3and its absence activate SGK3. If that is the case put an arrow from Mesothelin to DKK and show a downward arrow near DKK showing the down regulation and show SGK3 as activated SGK3 kinase.

RE: Figure 3 has been modified, as you suggests. Moreover in the text the sentence as modified

In the cytoplasm, FOXO3 is ubiquitinated and then degradated  in a proteasome-dependent manner

 Please,  see the following definition of Ubiquitination

“Ubiquitination (Ubiquitylation) is an enzymatic process that involves the bonding of an ubiquitin protein to a substrate protein. This has sometimes been referred to as the molecular “kiss of death” for a protein, as the substrate usually becomes inactivated and is tagged for degradation by the proteasome through the attachment of the ubiquitin molecule.”

The figure has been modified with arrows as you suggest

Thank you again for your suggestion

Round 4

Reviewer 3 Report

thank you very much for your additional  advices and recommendations to improve our manuscript

I have suggested authors to make concrete paragraphs so that short paragraphs with one or two sentences could be avoided. For that request, authors have now put everything as one paragraph. Looks like authors are not understanding how to structure solid paragraphs. When I suggested not to make one line and two lines as paragraphs that does not mean bring everything into some two page as one paragraph. It is beyond the scope of reviewer to pinpoint such basic structuring. As such I do not see any new information in this review and was conveyed during my initial review.